# Cognitive Tic-Like Phenomena in Gilles de la Tourette Syndrome

**DOI:** 10.3390/jcm10132749

**Published:** 2021-06-22

**Authors:** Piotr Janik, Anna Dunalska, Natalia Szejko, Andrzej Jakubczyk

**Affiliations:** 1Department of Neurology, Medical University of Warsaw, Banacha 1a Str., 02-097 Warsaw, Poland; piotr.janik@wum.edu.pl (P.J.); natalia.szejko@wum.edu.pl (N.S.); 2Department of Bioethics, Medical University of Warsaw, Żwirki i Wigury 63 Str., 02-091 Warsaw, Poland; 3Department of Psychiatry, Medical University of Warsaw, Nowowiejska 27 Str., 00-665 Warsaw, Poland; ajakubczyk@wum.edu.pl

**Keywords:** Gilles de la Tourette syndrome, cognitive tic-like phenomena, coprolalia, echophenomena, obsessions, anxiety disorder

## Abstract

Coprolalia and echophenomena repeated in the patients’ mind (CTPh—cognitive tic-like phenomena) have been rarely recognized as part of Gilles de la Tourette syndrome (GTS) symptomatology and their assignment to tics, OCD or other psychopathologies has not been settled. The aim of the paper was to assess the incidence and clinical associations of CTPh in GTS, and to establish if CTPh belong to the tic spectrum. We performed a prospective, one-registration study on a cohort of 227 consecutive patients with GTS. CTPh were diagnosed during the interview and defined as brief, sudden, involuntary thoughts that had corresponding complex vocal tics. CTPh occurred at some point in the lives of 34 (15.0%) patients. The median age at onset of CTPh was 14.5 years (IQR: 10.5–17.5). CTPh were found more frequently in adults, with the most frequent onset in adolescence (44.1%). Four mental phenomena resembling tics were recognized: echolalia (*n* = 17), coprolalia (*n* = 16), palilalia (*n* = 13) and repeating of words in the mind (*n* = 7). The older the age of patients, the more severe tics, and anxiety disorder significantly correlated with CTPh. CTPh may be considered as a part of tic spectrum with a substantial impact of anxiety disorder. CTPh are a late and age-related symptom of GTS.

## 1. Introduction

Gilles de la Tourette syndrome (GTS) is a neurodevelopmental disorder that most commonly affects children and adolescents, but in some cases persists into adulthood. According to the *Diagnostic and Statistical Manual of Mental Disorders* (5th ed., DSM-5) in order to diagnose GTS [1], numerous motor tics and at least one vocal tic must be present for at least one year and appear before the age of 18. Nevertheless, in approximately 85.7% of the patients with GTS, co-occurring psychiatric disorders are present [2]. Individuals with GTS often report the presence of obsessions and compulsions. In fact, obsessive-compulsive disorder (OCD) could be found in approximately 30% of GTS patients and is, therefore, one of the most frequently encountered comorbidities [3]. Nevertheless, based on clinical practice, we observed that these patients additionally tend to experience other mental phenomena such as using foul language in thought, mentally repeating their own words (palilalia) and echoing in the mind expressions heard during conversations or while watching television (echolalia). Repeated phrases of this nature can appear as either short wording or longer utterances, such as a sentence or part of a statement. Moreover, patients with GTS seem to evince a certain disposition to count objects in their surroundings, for instance, books on the shelf, corners of paintings, numbers of license plates, etc. According to O’Connor [4], this kind of mental phenomena can be defined as cognitive tics and described as thoughts, phrases, urges, songs, words or scenes that intrude into the consciousness of the patient and are perceived as difficult to remove.

On the basis of current understanding, tics are sudden, rapid, recurrent, nonrhythmic motor movements or vocalizations [5]. Even though this kind of description corresponds well with clonic tics, which are brief, fast, jerk-like movements [6], there are some tics that do not fulfil the above criteria and therefore should be distinguished. Longer and slower movement is typical for dystonic tics [7], whereas tonic tics are contractions of a muscle group and they are devoid of the movement effect or accompanied by only slight visible motion [8]; finally, blocking tics are a presentation of a rapid cessation of muscle activity and voluntary action, e.g., walking or speech [9,10,11]. According to current definition, tics are repetitive behaviors, not repetitive thoughts; hence, cognitive tic-like phenomena (CTPh), which occur exclusively in the thoughts of the patient, do not fulfil the diagnostic criteria for tics.

It is crucial to differentiate CTPh with the spectrum of obsessive-compulsive phenomena. OCD can be diagnosed when a patient reports obsessions/compulsions or both, the presence of which is time consuming or debilitating in terms of daily functioning and cannot be attributed to physiological effects of a substance or another medical condition. Obsessions are defined as recurrent, persistent thoughts, urges or impulses that are perceived by the patient as intrusive and unwanted; therefore, they are usually associated with anxiety and distress. In contrast to obsessions, CTPh are experienced as pleasant or neutral, do not lead to negative repercussions and are sequences in themselves (one type of CTPh does not connect to another phenomenon), while obsessions are often a coherent line of thoughts [4,12]. The differentiation with mental compulsions, such as silent checking or praying every time a patient has a bad thought, is even more challenging. Mental compulsions are performed by a patient as a response to obsessions or as an expression of the urge to apply self-imposed rigid rules, are driven to perform covert acts in thoughts that may be recognized by patients as senseless, excessive, difficult to resist and anxiety related until the act is completed. In contrast to compulsions, CTPh are not performed to reduce mental distress or prevent some dreaded situations. They are rather of a pleasant or stimulating nature. Obsessions and compulsions are also consistent and specific for a particular patient and do not tend to alter significantly in time, while CTPh can be effectively substituted by a competing stimulation [12]. However, the differential diagnosis between tics and the symptoms of OCD phenomenology can cause problems in clinical practice. Ganos et al. [12] mention the term ‘tic-like obsessions/compulsions’ and describe it as intermediate phenomena between tics and obsessions. Other comorbid disorders manifested by difficulties in thought control (i.e., depressive or anxiety disorder, attention-deficit/hyperactivity disorder, ADHD) or mental stereotypies, such as stereotyped behavior in autism spectrum disorder (ASD), should be taken into consideration in differential diagnosis and considered as possible clinical correlates of CTPh. However, these associations are yet to be thoroughly studied and the assignment of CTPh to tics, OCD or other psychopathologies has not been settled. Based on our clinical experience, we suspected that long-lasting thoughts may be closer to OCD phenomenology, while brief, sudden, involuntary thoughts, especially those analogous to typical vocal tics, such as echolalia, palilalia and coprolalia, may be rather a part of the tic spectrum. Therefore, so as not to prejudge the assignment of these mental acts to the specific symptom group, we decided to use the term cognitive tic-like phenomena (CTPh) in our study.

Psychopathology related to GTS may be different in children and adults. Hirschritt reported that adults and adolescents were most likely to have OCD as well as mood, anxiety, eating and substance use disorders, whereas children were more likely to have ADHD [2]. This latter observation stays in line with the previous finding that up to 85% of the ADHD patients report symptoms’ remission in adulthood [13]. On the other hand, some patients, prospectively reporting the occurrence of CTPh, may not have developed the symptoms of a given comorbid disorder yet. Previous research has demonstrated that ADHD symptoms began 1–3 years prior to tic onset [2,14], OCD began 1–6 years after the first tic onset [2,15], while mood and substance use disorders began even later (13 and 16 years, respectively) [2]. These results suggest that psychiatric comorbidities may appear at any time during the course of GTS including early childhood, adolescence and adulthood. Based on these observations, we suspected that clinical characteristics of CTPh may be different in children vs. adults.

To the best of our knowledge, there have been no studies that investigate CTPh in a clinical sample. The aims of this study were (i) to assess the prevalence and age at onset of CTPh in individuals with GTS; (ii) to assign this phenomenon to tics, OCD or another psychopathology; (iii) to distinguish between the clinical correlates of CTPh for children and adults. Based on available findings as well as our clinical experience, we hypothesized that (i) the prevalence of CTPh may vary depending on the age of the patients, (ii) CTPh are part of tic phenomenology and are neither part of OCD nor another psychopathology and (iii) clinical correlates of CTPh in children differ from those in adults.

## 2. Materials and Methods

### 2.1. Study Participants

The cohort of individuals with GTS comprised 241 consecutive patients, who were evaluated from 2013 to 2019. In 14 patients, data on the presence of CTPh were not available, as these were young children unable to comprehend questions during the interview, resulting in a total sample size of 227 patients. In this group, the age of the patients ranged from 5 to 50 years (median, IQR: 13, 10–22.5 years; 179 males, 78.9%). In total, 140 children (61.7%, median, IQR: 10, 8.75–12 years; 111 males, 79.3%) and 87 adults (median, IQR: 25, 21–32 years; 68 males, 78.2%) were enrolled. The median age of tic onset was 6 years (IQR: 5–7 years, range: 2–17). The median duration of GTS was 4 years (IQR: 3–7 years, range: 1–13) in children and 18 years (IQR: 12–23 years, range: 6–39) in adults. Overall, 189 (83.3%) patients had at least one psychiatric comorbidity. Psychiatric comorbidities of GTS patients are presented in Table 1.

In total, 103 patients already received therapy at the time of examination, including behavioral therapy, tic-reducing drugs and OCD symptom-relieving medications.

### 2.2. Procedures

All the patients were recruited from one single outpatient clinic and were personally reviewed and evaluated by the same clinician, who was well experienced in tic disorders (PJ). Patients were referred to the clinic by general neurologists and psychiatrists, due to problematic diagnosis or tics refractory to treatment, or sought medical advice on their own because of troublesome tics. The study was designed as a one-time registration study, as patients were registered in the database only once, and no additional clinical data obtained in follow-up visits were included in the analysis.

The patients were evaluated for the clinical diagnosis of GTS according to the Diagnostic and Statistical Manual of Mental Disorders criteria that were valid at the time of evaluation (DSM-IV-TR, DSM-5). All patients were systematically interviewed with the aid of a semi-structured interview that comprised demographic and clinical data that was gathered for all the patients. This schedule is based on the Tourette syndrome international database consortium (TIC) data entry form, developed by Freeman et al. [16], in which the investigator (PJ) participated and subsequently used this form in clinical practice. This interview was slightly modified over time and expanded with the questions on different types of tics including cognitive tic-like phenomena, coprolalia, echolalia and palilalia (each symptom was scored as either present or absent).

The prevalence of the most common comorbid disorders encountered in GTS was evaluated on the basis of the same, mentioned above, semi-structured clinical interview. Disorders that were listed in this semi-structured interview were as follows: ADHD, OCD, depression, anxiety disorder (including different forms of anxiety disorders: phobias, panic disorders, generalized anxiety disorder and separation anxiety disorder), oppositional defiant disorder, conduct disorder. The list of obsessions and compulsions included in the Yale–Brown Obsessive-Compulsive Scale (Y-BOCS) was used to establish clinical spectrum of OCD. Each patient was carefully questioned about all the symptoms included in the DSM as the diagnostic criteria of the above-mentioned comorbid disorders. The diagnoses of mental disorders that was made in psychiatric clinics before our evaluation were accepted and included into the analyses. One-third of children and adolescents, with more complex psychopathology, were assessed with the M.I.N.I. International Neuropsychiatric Interview for Children and Adolescents. Patients with severe psychiatric comorbidities were referred to psychiatrist to confirm the diagnosis. In this case, the psychiatric diagnosis was considered to be definite, and as such taken into consideration in the study analyses.

Tic severity was measured using the Yale Global Tic Severity Scale (YGTSS) on the day of each patient’s evaluation [17]. CTPh were not evaluated with YGTSS, as they are not included in this assessment tool. In contrast to children and adolescents, in whom most clinical information was provided by their parents, adults reported the information themselves. All questions asked during the interview were part of the routine practice and, therefore, no refusal rate is reported in this study.

### 2.3. Definition and Differential Diagnosis of Cognitive Tic-Like Phenomena

CTPh were identified during the dialogue-based interview through an active inquiry. Due to lack of instruments validated for the assessment of CTPh, we used the clinical definition established by previous investigators [4], which was modified by the author of the study (PJ) with regard to tic disorders. In this study, only one examiner (PJ) performed evaluation with regard to CTPh and the diagnosis of these kinds of thoughts was not confirmed by other researchers. As CTPh were defined as a mental (i.e., internal) act, their assessment was based entirely on the individual’s retrospective account. The characteristics of CTPh including the type, age of onset and persistence during evaluation were included in the questionnaire, which we used to obtain demographic and clinical data.

CTPh were defined as brief, sudden, recurring, neutral or pleasant mental acts, which had the equivalent among typically recognized complex vocal tics such as coprolalia, echolalia, palilalia and repeating of words or phrases. Complex and long-lasting thoughts, e.g., mentally jumping over telegraph poles along the roadside, drawing three-dimensional constructions in the mind or having the same tune in the mind for hours, were not classified as CTPh, as they do not have an analogous vocal tic. This type of diagnostic attitude precluded confusing CTPh with delusional disorder, ruminations in depression and obsessions, hyperquantivalent ideas or other psychiatric disorders. Importantly, none of the patients included in the study group were diagnosed with a psychotic disorder or bipolar disorder (Table 1).

CTPh were differentiated with obsessions and mental compulsions according to the differential diagnosis described in the introduction. For this purpose, interviews with patients covered questions about their emotional attitude towards appearing thoughts, the consequences of the thoughts, the natural form of the thoughts’ sequence, the possible aim of the thoughts, the resistance the patient made against these thoughts and if they were acceptable or not for the subject. Patients received questions on whether they had ever experienced short-lasting thoughts that appeared spontaneously in the mind. We asked the patients if they had repeated their own words or phrases in the mind two or more times in a row (mental palilalia), if they had imitated several times in their mind words or phrases spoken by other people (mental echolalia) or if they had repeated curse words voicelessly (mental coprolalia). We also asked the patients when those thoughts started, if they were present at evaluation and if the patients had ever experienced typical vocal coprolalia and echophenomena, to assess the relationship between CTPh and their vocal equivalent. If the patient reported the presence of at least one CTPh during the lifetime, he/she was included into CTPh+ group. The CTPh- group consisted of patients who had never experienced thoughts of this kind.

### 2.4. Lifetime vs. Current Analyses

We assumed that the factors that influence the lifetime prevalence of CTPh in GTS may differ from those that are related to CTPh at the time of examination. For example, a lifetime diagnosis of OCD does not necessarily contribute to the occurrence of CTPh as both symptoms could have occurred at different moments in the lifetime. In order to assign CTPh to tics or OCD phenomenology, lifetime and current comparative analyses were conducted. The lifetime comparative analysis included the group of the patients without any history of CTPh (CTPh-) and the group of the patients with a history of CTPh in the past or at the time of evaluation (CTPh+). When categorizing patients as having CTPh at evaluation, we took into account the two preceding weeks. In the current comparative analysis, we divided all the patients in two groups: CTPh current+ (GTS patients with CTPh at the time of evaluation) and CTPh current- (GTS patients without CTPh within last two weeks). As a result, CTPh current- included both patients without any history of CTPh and those with CTPh in the past but not at the time of evaluation. Moreover, in this analysis, we took into account only CTPh and OCD symptoms that were present at the time of the examination. The exact relationship could be determined only when both symptoms were present at the same time.

### 2.5. Statistical Analysis

Our plan of analysis included a comparison of the CTPh+ and CTPh- groups when considering the demographics and distribution of psychiatric comorbidities as well as premonitory urge and tic severity. Apart from analysis in the whole group, we also conducted the same comparison in adults and children. The statistical analyses were performed using STATISTICA version 13.1 (StatSoft, Tulsa, OK, USA), SPSS version 25 software (IBM, Armonk, NY, USA) and R programming language. The Shapiro–Wilk test was used to assess the normality of distribution. In the case of the parametric variables, data were presented as arithmetic means and standard deviations (mean ± SD). For the non-parametric variables, we chose the median and quartiles (25:75) to present data. The categorical variables were presented as frequencies (percentages). Parametric data was compared using an independent t-test. For the nonparametric data, we used the Mann–Whitney U-test. The Chi-square test was performed for comparison of categorical data. For comparison of subgroups of patients with different CTPh, we used a test for small groups, the Fisher’s exact test.

In both analyses, the *p*-value had to reach <0.05 for the comparisons between the groups to be considered significantly different. In the course of the primary analyses, we isolated variables that emerged as significant and entered them into a logistic regression analysis in which the dependent variable was the presence of CTPh. Additionally, sex and age served as control variables in the multivariate model. Logistic regression analysis was also used to check whether CTPh were associated with their vocal equivalent.

## 3. Results

### 3.1. General Characteristics

All the patients who experienced CTPh recognized them as neutral, neither pleasant nor distressful in any way and reported them only when they had been asked for, and never spontaneously. Some individuals did not even realize they had these mental thoughts. CTPh occurred at some point in the lives of 34 (15.0%) patients. Four types of mental phenomena were evaluated: cognitive echolalia (*n* = 17), cognitive coprolalia (*n* = 16), cognitive palilalia (*n* = 13) and repeating words or phrases in the mind (*n* = 7). The presence of one CTPh occurred in 22 patients, of 2 CTPh in 5 patients and of 3 CTPh in 7 patients. The group of patients with only mental coprolalia (*n* = 8) was compared with the group with only mental echophenomena (palilalia or echolalia or both, *n* = 11). These groups did not differ with regard to age, gender, YGTSS, premonitory urges, possibility of control over tics, relief after tic execution and psychiatric comorbidities. In 7 patients, mental coprolalia co-occurred with at least one echophenomena.

CTPh were reported in both patients who experienced their vocal equivalent as well as in those who did not present analogous vocalization. We checked whether CTPh were associated with their vocal equivalent but failed to demonstrate such relationship. Moreover, we assessed whether mental echophenomena had the tendency to occur together and demonstrated that mental palilalia and echolalia were associated with each other (*p* < 0.000001, OR 19.56, CI 95%: 5.47–77.82). In contrast, the presence of mental coprolalia did not increase the risk of mental echolalia (*p* = 0.086, OR 3.43, CI 95%: 0.78–13.61) or mental palilalia (*p* = 0.138, OR 3.16, CI 95%: 0.65–14.11).

In the univariate analysis, CTPh were associated with patients’ older age at evaluation, greater severity of tics, premonitory urges and anxiety disorder (Table 2). Neither lifetime OCD (Table 2) nor current OCD nor past OCD (*p* = 0.2449 and *p* = 0.8048, respectively) were associated with CTPh. The YBOCS score that was available for 35 of 47 patients did not differentiate the CTPh+ and CTPh- groups (Table 2).

CTPh were associated with older age of the patients, greater tic severity and anxiety disorder in logistic regression analysis (Table 3).

### 3.2. Children/Adolescents vs. Adults with CTPh

The prevalence of CTPh differed between adults and children/adolescents: 23.0% (20/87) and 10.0% (14/140), respectively (*p* = 0.016). The age at onset of CTPh was known in 26 of all cases with the median value of 14.5 years (IQR 10.5–17.5). In 7 of the cases, CTPh started during childhood (before 11 years of age), in 15 of the cases it started during adolescence (12–18 years of age) and in 4 of the cases it started in adulthood.

In the children/adolescent group, variables which, in univariate analysis, were significantly associated with CTPh and were later included in the logistic regression model were as follows: age at evaluation and YGTSS. Both variables confirmed their significant association with CTPh in children/adolescents when multivariate regression model was tested. In the adult group, no significant associations for CTPh were found (Table 2 and Table 3).

## 4. Discussion

In the first step, in order to recognize CTPh in GTS and to avoid overlapping with other cognitive acts, we differentiated them from obsessions and mental compulsions. Based on our clinical experience and according to the definition of O’Connor [4], we recognized CTPh as short-lasting (not time consuming), appearing in mind suddenly/more automatically (not intrusive) and as a pleasant or neutral phenomena (not distressful). In case of our study, all the patients who experienced CTPh recognized them as neutral, not because they appeared suddenly in the mind, but due to patients’ emotional attitude to these mental acts and lack of interfering with normal living. Importantly, e.g., obsessions may happen suddenly, but they are not neutral because they are time consuming and disturb daily functioning. Finally, we focused on thoughts that were analogous to typical vocal tics, excluding those that did not correspond with vocal tics.

We found that cognitive tic-like phenomena occurred in 15% of the patients. Mental echophenomena were strongly correlated with each other except for mental coprolalia. This suggests that different CTPh may be related and appear together or occur regardless of each other. There is a possibility of CTPh occurrence in both patients suffering from the vocal equivalent as well as in those who do not present symptoms of this kind. It means that both variants, vocal and cognitive, of a given tic do not necessarily appear in one particular individual. Echophenomena (pali- and echolalia) and coprolalia were similarly presented either in a form of vocalization or mental act. These observations indicate only a certain degree of relationship between the mental and vocal forms of the tic.

We also looked for potential differences between various types of CTPh. However, we did not find any significant demographic and clinical differences between mental coprolalia and mental echophenomena, although very low numbers of comorbid psychiatric disorders and other variables taken into the comparative analysis do not allow for any final conclusions to be drawn. On the other hand, in a few cases mental coprolalia and mental echophenomena appeared in the same patients, which would be rather expected if these CTPh were to have a similar background. These statements should be verified in future studies.

As we mentioned, the emotional attitude of the patients toward the experienced mental acts was neutral and not troublesome in any way, and they reported these thoughts only when they had been actively inquired, never spontaneously. That same was true for mental (silent) coprolalia. It was somehow surprising because, e.g., vocal coprolalia nearly always causes distress. Based on clinical experience, the possible explanation is that ‘loud’ coprolalia is distressful because it draws attention of other people and is socially unacceptable. If vulgar words/sentences are distorted or uttered in a low voice they are less bothersome and cause less distress for a patient. That is why we consider CTPh egosyntonic symptoms.

CTPh correlated significantly with tic severity, premonitory urges and anxiety disorder. We formulated a hypothesis that particular mental phenomena reported by certain GTS patients may be a form of tic manifestation and we juxtaposed it with the theory that these symptoms are a representation of another psychopathology. Inferring from the positive correlation between CTPh and YGTSS (Table 2 and Table 3), we consider that higher tic severity may increase the risk of CTPh appearance. The opposite situation, when CTPh add significantly to the impairment caused by tics, is less likely because CTPh themselves are perceived by patients as egosyntonic; hence, they do not require additional or more intensive treatment. Additionally, correlation between premonitory urges and CTPh, which emerged statistically significant in univariate analysis (Table 2), provides further evidence validating the hypothesis of CTPh affiliation to the tic spectrum. On the other hand, it seems that CTPh are not premonitory urges themselves. Premonitory sensations precede tics and disappear temporarily after a tic is executed. A relief after a tic has been performed is often associated with vanishing of these unpleasant urges. We did not show in our study the significant associations of CTPh with these tic-related features i.e., premonitory urges (Table 3) and relief after tic execution (Table 2). Moreover, based on the definition of CTPh that we adopted from the work of O’Connor [4], these mental acts are sequences in themselves, which means that a CTPh does not precede a typical tic (motor or vocal) and does not lead to another mental phenomenon. Based on the association of CTPh with tic severity and urges preceding tics, we are prone to classify the examined phenomena as tic spectrum symptoms rather than recognize them as a manifestation of comorbid psychopathology. Nevertheless, the association of CTPh to the tic spectrum should be made with caution. First, we did not find any association of CTPh with typical tic-related features such as possibility of control over tics and relief after tic execution (Table 2). Secondly, the correlation between the CTPh and anxiety disorder was exposed (Table 2 and Table 3). The latter correlation is not easy to explain. One of the reasons for that could be the close relation of anxiety and OCD, as anxiety is a typical consecution of obsession and often a causative factor of compulsion. However, we did not find any correlations between OCD (lifetime, current or past) and CTPh in our study, although we were not able to establish (due to recall bias) if CTPh were present at the same time as OCD symptoms in the past. This could have affected the reliability of the analysis of relationship between past OCD and past CTPh. Moreover, the YBOCS score did not differentiate CTPh+ and CTPh- groups either. This kind of result could arise from the fact that CTPh were defined strictly in the context of tic phenomenology, not the definition of obsessions and mental compulsions. Nevertheless, the lack of association between CTPh and OCD was surprising. The fact that the age of CTPh onset falls in most patients in early adolescence, a period when tics begin to subside and after the peak tic severity, which is usually seen between 10–12 years [18], also suggests that tic severity does not have to be the crucial factor regarding CTPh appearance.

Moreover, we hypothesized that patients’ age could be a variable associated with the rate of CTPh, especially in patients below the age of 18. Age of the CTPh onset falls during adolescence in most patients, although it may occur at any time during the course of the disease. There is an evident association between the prevalence of this mental phenomenon and the age of the patients (Table 2 and Table 3). The fact that CTPh appear several years after the development of the first tic suggests that most individuals do not present with CTPh. We also found that CTPh occurred in adult patients twice as often as in children. It is possible that younger patients actually report CTPh to a lesser extent than older individuals, much like premonitory urges [19,20], or that CTPh may have natural tendency to manifest over time. However, it may also indicate difficulty in obtaining information from younger children and that the diagnostic material collected from children is less reliable because of the children’s incomprehension of the questions during the interview and the lack of parents’ insight into child’s mental state. Taking into consideration the fact that CTPh were never reported spontaneously by patients and were realized only due to active inquiry, proper cooperation and communication appear essential in diagnosing these phenomena. The acknowledgment of this may contribute to the understanding why the issue of CTPh is still not sufficiently examined as the majority of studies in GTS involve the child population.

We found significant differences in correlates of CTPh in children/adolescents and adults. The older the child/adolescent patient was, the greater risk of CTPh they had. This correlation was not observed in adult group (Table 2 and Table 3). Moreover, in contrast to adults, the risk of CTPh occurring in children and adolescents was associated with tic severity, which suggests that CTPh appear generally in children and adolescents with more severe tics. These differences between children and adult patients with GTS can provide with noteworthy conclusions regarding the pathogenesis of these symptoms. Taking into consideration the fact that certain co-occurring disorders have the tendency to be present in the specific age groups (Table 1), we can speculate that the whole group of symptoms, depending on the age of the patient, can be a factor related to CTPh occurrence. In this case, CTPh would have various causes, distinct for age groups and specific for particular patients. Thereby, the differences in clinical picture of children and adults provide further evidence that pathogenesis of CTPh is complex, depends on multiple factors and is related to the phenotypic variability of GTS.

Another possible consideration of CTPh character arises from the careful understanding of its definition: CTPh are pleasant or neutral, stimulating in their nature rather than regulating negative emotional states. Current research emphasizes two main basic human needs that should be taken into general consideration in describing and understanding psychiatric phenomena: the regulation of negative emotional states and reward seeking (stimulation) [21,22]. Importantly, these needs may be addressed with different strategies (mature or immature, short- or long-term, conscious or unconscious and constructive or destructive strategies). A commonly used regulative/reward-seeking strategy is the use of psychoactive substances, for instance, alcohol or stimulants. It can be very carefully speculated that CTPh may in some cases act like an internal, immature mechanism with an auto-stimulating function [4]. The similar age at onset of substance use disorders [2] and CTPh found in our study (median: 16 and 14.5, respectively) can possibly serve as the evidence encouraging this hypothesis. The prevalence of alcohol or substance use disorder was not assessed in our study, yet it would be interesting to address this issue in future studies and to investigate whether the presence of CTPh may serve as protector or rather as a risk factor of substance use in GTS and other clinical and non-clinical samples.

The value of the performed study arises from both extending of scientific knowledge in the topic of mental phenomena in patients with GTS and the prospective clinical application of acquired data. In opposition to OCD, CTPh do not require treatment as they are perceived by patients as neutral and egosyntonic. It also seems important to pay attention to the presence of mental phenomena resembling tics, which do not fulfill the current diagnostic criteria of this hyperkinetic movements and, although unobservable, may belong to the spectrum of tic phenomenology. Our results suggest that GTS-affected patients who also have CTPh at the time of examination should be carefully evaluated for existing anxiety disorders and more severe tics. This could certainly influence the choice of appropriate treatment.

There are several limitations to our study which include the following: lack of a control group; lack of any validated instrument in CTPh assessment; no confirmation of CTPh diagnosis from the another researcher who would evaluate the patient’s report independently; the established CTPh definition was limited to tic phenomenology with exclusion of overlapping, long-lasting mental phenomena from the OCD spectrum; recall bias could potentially influence the reported prevalence rate and age at onset of CTPh; the one-time registration study design could influence the rate of CTPh and psychiatric comorbidities; no validated rating scales for all adults and some patients under the age of 18 were used for the assessment of comorbidities; no rating scale was used to assess premonitory urges quantitatively; there could have been referral bias as the patients were evaluated by a neurologist and the cases with more severe psychopathology were referred to psychiatric clinics, which explains the low numbers of OCD and depression in the children group.

## 5. Conclusions

We conclude that some complex tics may appear in two forms, vocal and mental. CTPh should presumably be considered as a part of the tic phenomenology with a crucial impact of comorbid anxiety disorder. CTPh are rather rare and late symptoms of GTS, tend to appear over the course of the disease and, hence, are an age-dependent symptom.

## Figures and Tables

**Table 1 jcm-10-02749-t001:** Psychiatric comorbidities in children, adolescents and adults with GTS.

Comorbid Mental Disorder	Children (*n* = 95)	Adolescents (*n* = 44)	Adults (*n* = 88)
ADHD	30.5%, *n* = 29	20.5%, *n* = 9	20.5%, *n* = 18
OCD	6.3%, *n* = 6	18.2%, *n* = 8	37.5%, *n* = 33
Depression	0%, *n* = 0	6.8%, *n* = 3	33.0%, *n* = 29
Anxiety disorder	44.2%, *n* = 42	61.4%, *n* = 27	51.1%, *n* = 45
ODD	25.3%, *n* = 24	11.4%, *n* = 5	3.4%, *n* = 3
ASD	3.2%, *n* = 3	6.8%, *n* = 3	2.27%, *n* = 2

GTS—Gilles de la Tourette syndrome, ADHD—attention deficit hyperactivity disorder, OCD—obsessive-compulsive disorder, ODD—oppositional defiant disorder, ASD—autism spectrum disorder. Groups are defined as children (aged 5–11), adolescents (aged 12–18), adults (aged > 18).

**Table 2 jcm-10-02749-t002:** Comparison of lifetime CTPh- and CTPh+ groups.

	All GTS Patients (*n* = 227)			Children and Adolescents with GTS (*n* = 140)			Adults with GTS (*n* = 87)		
	CTPh+ (*n* = 34)	CTPh- (*n* = 193)	*p*	CTPh+ (*n* = 14)	CTPh- (*n* = 126)	*p*	CTPh+ (*n* = 20)	CTPh- (*n* = 67)	*p*
Age at evaluation [years] [median] (IQR)	18 (11)	12 (12)	**0.001**	11.5 (3.75)	10 (3.75)	**0.004**	25 (11.75)	25 (11)	0.279
Sex (male/female)	29//5	143/50	0.4416	13/1	95/27	0.335	16/4	52/15	1.0
YGTSS [median] (IQR)	63.5 (38)	43 (33)	**0.003**	64.5 (28.5)	37 (33.75)	**0.012**	60.5 (45.5)	51 (24)	0.3585
Premonitory urges	*n* = 28 (82.4%)	*n* = 105 (54.4%)	**0.013**	*n* = 10 (71.4%)	*n* = 55 (45.1%)	0.2	*n* = 18 (90%)	*n* = 49 (73.1%)	0.238
Possibility of control over tics	*n* = 27 (79.4%)	*n* = 149 (77.2%)	0.7613	*n* = 10 (71.4%)	*n* = 86 (61%)	0.8906	*n* = 17 (85%)	*n* = 63 (94%)	0.2337
Relief after tic execution	*n* = 28 (82.4%)	*n* = 131 (57.7%)	0.2815	*n* = 10 (71.4%)	*n* = 74 (58.7%)	0.6698	*n* = 18 (90%)	*n* = 57 (85%)	0.9646
ADHD	*n* = 7 (20.6%)	*n* = 49 (25.4%)	0.701	*n* = 6 (42.9 %)	*n* = 31 (25.4%)	0.284	*n* = 1 (5%)	*n* = 17 (25.4%)	0.097
OCD	*n* = 9 (26.5%)	*n* = 38 (19.7%)	0.503	*n* = 3 (21.4 %)	*n* = 11 (9.0%)	0.326	*n* = 6 (30%)	*n* = 27 (40.3%)	0.568
YBOCS [median] (IQR)	20 (23)	20 (21)	0.6204	20 (20.5)	22 (24)	0.3518	19 (22)	19 (23)	0.8934
Depression	*n* = 8 (22.9%)	*n* = 24 (12.4%)	0.148	1 (7.1%)	2 (1.6%)	0.713	*n* = 7 (35.0%)	*n* = 22 (32.8%)	0.90
Anxiety Disorder	*n* = 24 (70.6%)	*n* = 90 (46.6%)	**0.017**	*n* = 10 (71.4%)	*n* = 57 (46.7%)	0.142	*n* = 14 (70.0 %)	*n* = 31 (46.3%)	0.108

CTPh—cognitive tic-like phenomena, GTS—Gilles de la Tourette Syndrome, YGTSS—Yale Global Tic Severity Scale, ADHD—attention-deficit hyperactivity disorder, OCD—obsessive-compulsive disorder, IQR—interquartile range. Lifetime prevalence of psychiatric symptoms and disorders is shown. After analysis of distribution, all continuous variables were found to have non-parametric distribution and therefore are presented as the median and interquartile range. Categorical variables are presented as frequencies (percentages). *p* < 0.05 is shown in bold. For the nonparametric data, the Mann–Whitney U-test was utilized. Chi-square test was performed for comparison of categorical data.

**Table 3 jcm-10-02749-t003:** Logistic regression analysis for the predictors of lifetime CTPh in GTS.

	All GTS Patients		Children with GTS	
	OR (95% CI)	*p*	OR (95% CI)	*p*
Age	1.04 (1.00–1.08)	**0.025**	1.306 (1.083–1.597)	**0.00631**
Sex	0.601 (0.196–1.528)	0.323	0.269 (0.015–1.443)	0.216
YGTSS	1.027 (1.010–1.046)	**0.00296**	1.037 (1.01–1.069)	**0.0101**
Premonitory urges	1.342 (0.879–2.28)	0.171	-	-
Anxiety disorder	2.94 (1.374–6.735)	**0.00719**	-	-

CTPh—cognitive tic-like phenomena, GTS—Gilles de la Tourette Syndrome, YGTSS—the Yale Global Tic Severity Scale, *p* < 0.05 is shown in bold. Analysis for adults with CTPh are not included since no association was demonstrated in univariate analysis.

## Data Availability

The datasets used and/or analyzed during the current study are available from the authors on reasonable request.

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
