# Peer review of "Cognitive Tic-Like Phenomena in Gilles de la Tourette Syndrome"

_jcm, 2021, doi:10.3390/jcm10132749_

Round 1
Reviewer 1 Report
This study focused on cognitive tic-like phenomena (CTPh), which were defined as brief, sudden, recurring mental acts, and had the equivalent among typically recognized complex vocal tics.
I believe that it provides an interesting perspective for deepening understanding of GTS.
However, it is questionable whether the method for assessing CTPh was robust enough.
In the limitations, the authors mentioned lack of any validated instrument in CTPh assessment, and it may not be easy to establish an assessment method for CPTh, which are newly proposed concepts. However, if the two researchers listened to the patient's statements and evaluated them independently, the reliability of CPTh assessment would have increased.
Is it correct to recognize that mental acts are always pleasant or neutral because they happen suddenly? Among complex vocal tics, coprolalia, unlike echolalia/palilalia, often causes destress with the recognition that the patient does not want to say it but does. The result that the presence of mental coprolaila did not increased the risk of mental echolalia or mental palilalia may correspond to this feature. Of 34 patients with CTPh, I would like to know what the difference was between 16 patients with mental coprolalia and 18 patients without mental coprolalia.
The authors stated in the limitations that there were no validated rating scales, and only the presence or absence of premonitory urges and obsessive-compulsive symptoms was used. If the rating scales were used to obtain dimensional data for them, it might have affected the results.
With increasing age, some patients may become less severe with their obsessive-compulsive symptoms and no longer meet the diagnostic criteria for OCD, while others may have more obsessive-compulsive symptoms instead of improvement of tics and meet the diagnostic criteria. Considering the above-mentioned possibilities, if there are data of past OCD diagnosis and/or lifetime OCD diagnosis in addition to current OCD diagnosis, it may be useful to add them to the analysis.
Author Response
Dear Reviewer,
Thank you for all the comments. We appreciate your suggestions. In the file below we address them and comment on the changes we introduced to the paper. Please see the attachment.

Reviewer 2 Report
This is a very interesting study on a phenomenon many clinicians seeing patients with TS have encountered, i.e., the presence of cognitive tics. The results presented certainly add to the unfortunately scarce literature on the subject and the authors must be complimented on this work.
Even though I am convinced that these (mostly complex vocal) phenomena correspond to tics (based on their semiology), I regret that the criteria we usually use for diagnosing tics have not been applied in the present: (i) the presence of premonitory sensations; (ii) possibility of at least partial control; and (iii) relief after tic execution. The authors state that these phenomena are egosyntonic and either neutral or pleasant, but this does not necessarily equate relief. Furthermore, my own clinical experience is not in accordance with this: some patients do indeed complain about these cognitive tics, feeling they are intrusive, distractive and time-consuming. Also, in the case of coprolalia, not egosyntonic at all… This is just a personal note but worth mentioning, I believe. Finally, it might be worth underlining that cognitive tics – as reflected by the data presented – do not represent premonitory sensations themselves.
Author Response

(The authors gave the same response as above.)

Round 2
Reviewer 1 Report
The manuscript has been carefully revised according to reviewer’s comments.
However, I felt that the description on pages 367 to 379 was a little difficult to understand. That is, based on the result that CTPh was not correlated with relief after tic execution, the authors stated that CTPh was not recognized as urges, while CTPh was not associated with typical tic related features. Please add an explanation on this point.
Author Response
Dear Reviewer,
We acknowledge your feedback. According to your suggestions we explained and clarified the paragraph between lines 367-375. We hope that now it is clear.